# Genome-Wide Identification and Expression Analysis of *BrGeBP* Genes Reveal Their Potential Roles in Cold and Drought Stress Tolerance in *Brassica rapa*

**DOI:** 10.3390/ijms241713597

**Published:** 2023-09-02

**Authors:** Ruolan Wang, Xiaoyu Wu, Ziwen Wang, Xiaoyu Zhang, Luhan Chen, Qiaohong Duan, Jiabao Huang

**Affiliations:** College of Horticulture Science and Engineering, Shandong Agricultural University, Tai’an 271000, China

**Keywords:** *Brassica rapa*, *GeBP*, expression profile, bioinformatics analysis, abiotic stress

## Abstract

The *GLABROUS1 Enhancer Binding Protein (GeBP)* gene family is pivotal in regulating plant growth, development, and stress responses. However, the role of *GeBP* in *Brassica rapa* remains unclear. This study identifies 20 *BrGeBP* genes distributed across 6 chromosomes, categorized into 4 subfamilies. Analysis of their promoter sequences reveals multiple stress-related elements, including those responding to drought, low temperature, methyl jasmonate (MeJA), and gibberellin (GA). Gene expression profiling demonstrates wide expression of *BrGeBPs* in callus, stem, silique, and flower tissues. Notably, *BrGeBP5* expression significantly decreases under low-temperature treatment, while *BrGeBP3* and *BrGeBP14* show increased expression during drought stress, followed by a decrease. Protein interaction predictions suggest that *BrGeBP14* homolog, *At5g28040*, can interact with DES1, a known stress-regulating protein. Additionally, microRNA172 targeting *BrGeBP5* is upregulated under cold tolerance. These findings underscore the vital role of *BrGeBPs* in abiotic stress tolerance. Specifically, *BrGeBP3*, *BrGeBP5,* and *BrGeBP14* show great potential for regulating abiotic stress. This study contributes to understanding the function of *BrGeBPs* and provides valuable insights for studying abiotic stress in *B. rapa*.

## 1. Introduction

*GLABROUS1 Enhancer Binding Protein (GeBP)* is a plant-specific DNA-binding protein initially discovered in *Arabidopsis thaliana*. GeBP and its homologs share two conserved regions: an unknown motif in the central region and a C-terminal hypothesized leucine zipper motif [1]. Both regions are crucial for downstream gene expression trans-activation.

At present, 16, 10, 10, 9, and 16 *GeBP* genes have been identified in *Arabidopsis thaliana* [1], *Solanum lycopersicum* [2], *Mangifera indica* L [3], *Glycine max* [4], and *Bambusoideae* [5], respectively. Previous studies highlight the importance of the *GeBP* gene family in plant growth and development. For instance, *GeBP* regulates trichome development through the expression control of the *GLABROUS1 (GL1)* gene [6]. In *A. thaliana*, *GeBP* also influences trichome elongation by modulating gibberellins and cytokinins in vivo [7]. Gilles Vachon et al. reported that *GeBP/GPL* redundantly contributes to the cytokinin hormone pathway. A mutant with triple loss-of-function of *gebp gpl1 gpl2* reportedly displayed reduced responsiveness to exogenous cytokinin in some cytokinin responses, including senescence and growth. Additionally, the triple mutant showed notably elevated levels of type-A ARR cytokinin response genes, suggesting that GeBP may enhance negative feedback regulation, leading to cytokinin insensitivity [6].

Several studies have demonstrated the crucial role of *GeBP* in responding to environmental stresses. Ray reported its upregulation under drought stress, similar to *B. napus*, indicating *GeBP’s* key role in plant response to drought stress [8]. In studies on apples, overexpression of *MdGeBP3* showed sensitivity to cytokinin, while its ectopic expression in *A. thaliana* reduced drought resistance [9]. Moreover, *GeBP* is vital in a plant’s ability to withstand the detrimental effects of heavy metals, like Cd, Cu, and Zn [10]. In response to adverse rhizosphere growth conditions, plants reorganize root architecture to avoid unfavorable areas. Recent studies have shown that GeBP-LIKE4 (GPL4) transcription factor is crucial in regulating root growth inhibition under heavy metal stress, particularly Cd stress. GPL4 is essential in the avoidance response to heavy metals, which occurs by regulating reactive oxygen species (ROS) concentration to inhibit root growth during heavy metal exposure. Notably, GPL4 also exhibits a similar response to excess Cu and Zn [10]. These findings emphasize the significance of *GeBP* and its associated transcription factors in mediating plant responses to heavy metal stress.

*Brassica rapa*, an important globally consumed vegetable in the Brassicaceae family, faces various stresses during growth. However, no studies have explored the *GeBP* transcription factor gene family in *B. rapa*. This study identifies the *BrGeBP* gene at the genome-wide level and analyzes its sequence characteristics, gene expression, regulatory mechanism, and epigenetics. The results provide insights for future studies on abiotic stresses of *BrGeBP* genes and offer genetic resources for biological breeding.

## 2. Results

### 2.1. Identification, Physicochemical Characterization of GeBP Family Genes

A total of 20 *BrGeBP* genes were identified from the *B. rapa* genome using the BLAST program with the *AtGeBP* gene sequence. These genes were renamed *BrGeBP1–BrGeBP20* and divided into four subfamilies. The physicochemical properties of BrGeBPs are presented in Table 1. *BrGeBP* genes are distributed across 6 of the 10 chromosomes, with chromosome 4 having the highest number (6), followed by chromosome 1 (5), and chromosomes 6 and 9 with 2 genes each. Chromosome 5 has the least representation, with only one gene. The *BrGeBP18–BrGeBP20* genes were found on the scaffold. Subcellular localization analysis revealed that all *BrGeBPs* are localized to the nucleus. Bioinformatics analysis of the 20 members showed varying amino acid lengths, molecular weights (MWs), and isoelectric points (pIs). The lengths of BrGeBP proteins range from 299 aa (BrGeBP10 and BrGeBP11) to 640 aa (BrGeBP9), molecular weights range from 33 kDa (BrGeBP10) to 73 kDa (BrGeBP9), and isoelectric point (pI) ranged from 4.56 (BrGeBP7) to 8.98 (BrGeBP9).

### 2.2. Phylogenetic Relationships

Phylogenetic analysis of BrGeBP proteins, along with *A. thaliana* and *O. sativa,* was carried out using the maximum likelihood method (Figure 1). The 51 *GeBPs* were clustered into 4 major groups (I, II, III, IV). The results from the evolutionary relationship showed that Group I was the largest, comprising 28 members, including *B. rapa* (6), *A. thaliana* (6), and all *OsGeBPs*. Group II comprised 13 members: *B. rapa* (7) and *A. thaliana* (6). Group III had four GeBP members: *B. rapa* (3) and *A. thaliana* (1). Group IV contained five members: *B. rapa* (3) and *A. thaliana* (2). Thus, *BrGeBPs* exhibited a closer relationship with *A. thaliana* than *O. sativa*.

### 2.3. Gene Structure and Conserved Motif Analysis

The phylogenetic relationship of 20 *BrGeBP* genes was constructed using MEGA-X, resulting in four subgroups based on their similarity levels. BrGeBP proteins exhibited three to nine presumed conserved motifs (Figure 2A), with all four subgroups showing similar motif organization and composition. Notably, *BrGeBP4, BrGeBP7,* and *BrGeBP13* had three motifs, while *BrGeBP9* had nine motifs. Motifs 1, 2, 3, and 4 were widely present in all *BrGeBPs*, indicating their conserved nature and potential importance in target gene regulation [11].

Functional domain prediction revealed that the four conserved motifs were situated within the DUF573 domain. Among the BrGeBPs, 14 possessed or exclusively had the DUF573 domain, making it the core domain of the GeBP family (Figure 2B).

The exon–intron configurations of *BrGeBPs* genes were examined to gain insights into the structural evolution of *BrGeBP* genes (Figure 2C). The results revealed 20 *BrGeBP* genes, with exon numbers ranging from 1 to 8. Among them, 13 had no introns, and *BrGeBP7* gene had 7 introns. Genes with similar structures were found clustered together.

### 2.4. Analysis of Promoter Cis-Elements of BrGeBPs

Promoters are crucial sequence elements in genes, initiating gene transcription. To explore the possible regulation mode of the *GeBP* genes, we analyzed the 2 kb upstream region of *BrGeBP* coding sequences for *cis*-elements prediction using PlantCARE (Figure 3). Light-responsive elements were the most abundant, averaging 10.4 elements per gene, significantly higher than other elements. *BrGeBP6* had the highest number, with 19 light-responsive elements. Except for *BrGeBP2* and *BrGeBP7*, all members contained anaerobic induction regulatory elements. Most genes contained various plant hormone response elements, with 75% having abscisic acid response elements and 45% containing MeJA-, salicylic acid-, and gibberellin-responsive elements. Additionally, ~60% of the genes had low-temperature response elements, indicating their potential roles in abiotic stress and hormonal responses. These promoter sequences also revealed other response elements, indicating functional diversity in the *GeBP* gene family.

### 2.5. Gene Expression Analysis of the BrGeBPs

#### 2.5.1. Analysis of Tissue-Specific Expression of *BrGeBPs*

The expression patterns of *BrGeBPs* in six organs and tissues (callus, flower, leaf, root, silique, and stem) were investigated based on public transcriptome data (Figure 4; Appendix A). Out of the 18 *BrGeBP* genes, all except *BrGeBP4* and *BrGeBP10* were expressed in at least 5 examined organs and tissues. Furthermore, tissue-specific expression was observed for some genes. For instance, *BrGeBP16* showed significantly higher transcript abundances in the callus, indicating its involvement in callus differentiation. *BrGeBP1* and *BrGeBP12* exhibited high expression in the silique, suggesting their critical role in fruit development. *BrGeBP3, BrGeBP6, BrGeBP8*, and *BrGeBP20* showed peak transcript levels in stems, indicating their roles in organ development and growth. *BrGeBP7* and *BrGeBP9* were highly expressed in flowers, indicating their importance in flower development. Notably, *BrGeBP12* was exclusively expressed in the silique, while *BrGeBP3* had a 12-fold higher expression in stems than in roots. Conversely, some genes, particularly *BrGeBP4* and *BrGeBP10,* showed no expression in any organ and tissue. These findings imply that *BrGeBPs* may be important during organogenesis, fruit development, and sexual reproduction.

#### 2.5.2. Expression Patterns in Response to Abiotic Stress Analysis

*B. rapa*, a winter storage crop, experiences both cold and drought stress. To assess this gene family’s role in stress response, we analyzed their gene expression profiles post-stress (Figure 5A–C; Appendix A). *BrGeBP5* and *BrGeBP17* showed higher expression under cold stress, while *BrGeBP3* and *BrGeBP14* exhibited higher expression under drought stress. We also examined lowly expressed genes in the transcriptome as they may play important roles in specific developmental stages, including *BrGeBP14,* and *BrGeBP17* for cold and drought stress treatments, respectively. The qRT-PCR-mediated functional validation of the key genes screened from the transcriptome confirmed that *BrGeBP5, BrGeBP14*, and *BrGeBP17* were significantly downregulated after cold treatment. Conversely, under drought stress (Figure 5C), *BrGeBP3* and *BrGeBP14* showed higher expression levels than the control, peaking at 4 h and 6 h, respectively.

### 2.6. Prediction of Protein–Protein Interaction

Proteins are pivotal in various cellular functions, interacting physically with molecules, like lipids, nucleic acids, and metabolites. Given the close evolutionary relationship between *B. rapa* and *A. thaliana*, we can predict the function of corresponding homologous genes in *B. rapa* through protein–protein interaction (PPI) analysis of the *GeBP* genes in *A. thaliana*. This study utilized the STRING database’s resources and algorithms to construct a predicted protein interaction network map for *AtGeBPs* (Figure 6A; Appendix A). STRING results identified significant partners of the *AtGeBP* gene, including *At2g25650* (GPL1) and *At2g36340* (GPL3). These genes interact with CPR5 and counteract its active role in cell expansion, thereby suppressing CPR5 in this process. The CPR5–GeBP interaction is crucial for plant bacterial resistance [12]. Additionally, we observed an interaction between *At1g11510* (homologous to *BrGeBP10*, *BrGeBP12*, and *BrGeBP15*) and DREB26 (*At1g21910*), a transcriptional activator involved in plant development and abiotic stress tolerance. DREB26 was found to participate in salt and osmotic stress response pathways [13] (Figure 6B). Furthermore, *At2g36340* (homologous to *BrGeBP7*) interacts with IAA27 (Figure 6C), a key regulator in various aspects of plant growth and development. Silencing IAA27 significantly affects the root system, leaf physiology, reproductive organs, and fruit quality [14]. Additionally, *At5g28040* (homologous to *BrGeBP3, BrGeBP14,* and *BrGeBP16*) interacts with DES1 and OASA1 (Figure 6D). DES1 is an essential enzyme involved in hydrogen sulfide (H_2_S) production, and ABA induction leads to sulfhydrylation modification of DES1 and H_2_S production, thus, promoting stomatal closure and conferring extreme temperature tolerance [15]. The ABA-regulated *OASA1* gene is induced in leaves, stems, and roots under high salt and heavy metal stress.

### 2.7. Prediction of microRNAs Targeting BrGeBPs

MicroRNAs are critical in post-transcriptional gene expression regulation, particularly in plant stress responses. To enhance our understanding of *BrGeBP* genes, we investigated microRNAs associated with these genes (Figure 7; Appendix A). Table 2 presents details of *BrGeBPs* and their respective targeted microRNAs. Eighteen microRNA types were identified to regulate *BrGeBPs*, such as Br-miR156a-3p, Br-miR156f-3p, Br-miR172d-3p, Br-miR172d-5p, Br-miR395a-5p, Br-miR395b-5p, Br-miR395c-5p, Br-miR395d-5p, Br-miR5711, Br-miR5714, Br-miR6032-3p, Br-miR9559-5p, Br-miR9565-5p, and Br-miR9569-5p, each targeting a single *BrGeBP* gene. Additionally, Br-miR159a, Br-miR5716, Br-miR5717, and Br-miR9555a-5p targeted two different *BrGeBP* genes. Notably, more microRNAs were found targeting a single *BrGeBP* gene than those targeting two genes. Among them, *BrGeBP4* emerged as the most targeted gene by microRNAs.

## 3. Discussion

Plants respond to biotic or abiotic pressures by regulating physiological and biochemical reactions, altering transcription factor expression to enhance stress resistance [1]. Prior research highlights *GeBP* transcription factors’ pivotal roles in plant growth, development, leaf senescence, and abiotic stress [3,10,16,17]. However, *GeBP* genes’ functional and family analysis in *B. rapa* remains unexplored. Here, we used bioinformatics to investigate *BrGeBP* gene family characteristics, gene expression, regulatory mechanisms, and epigenetics, facilitating future *BrGeBP* gene family functional studies.

This study identified 20 *B. rapa GeBP* gene family members. Based on phylogenetic relationships and sequence similarities, *BrGeBP* was classified into four groups. Interestingly, *B. rapa*, *A. thaliana*, and *O. sativa* belonged to different groups, indicating distant relations between *B. rapa* and *O. sativa*. Conversely, *A. thaliana* and *B. rapa* grouped together, suggesting they shared similar physiological functions for their *GeBP* genes. Motif designs were also similar within the groups, implying a potential role of *BrGeBP7* and *BrGeBP13* in regulating *B. rapa* trichome cell elongation.

Post-transcriptional gene regulation involves miRNAs, which are single-stranded non-coding microRNAs [18,19]. Br-miR395a-5p, Br-miR395b-5p, Br-miR395c-5p, Br-miR395d-5p, and Br-miR5717 were found to target *BrGeBP4* of the *GeBP* gene family. These microRNAs work together to regulate the expression of *BrGeBP4*, which is crucial for precisely controlling plant growth and development [1]. They may target different sites on the gene to achieve distinct regulatory effects. Studies show that miR395’s targeting of growth-regulating factor (GRF) transcription factors and the *Sulfate transporter 4;1* (*SULTR4;1*) gene in *A. thaliana* regulates sulfur (S) uptake and distribution [20]. This suggests that miR395’s regulation of *BrGeBP4* might influence S metabolism and stress response in *B. rapa*. Notably, we identified a novel microRNA, Br-miR5717, targeting *BrGeBP4*. To our knowledge, this is the first report of miR5717 targeting a *GeBP* family transcription factor. Thus, miR5717 may play a unique role in regulating *GeBP*-mediated gene expression in *B. rapa*. Altogether, the concurrent targeting of a *GeBP* family member by these five microRNAs achieves greater intricate and precise gene regulation, optimizing the function of the gene family member in *B. rapa* growth and development.

*BrGeBP5* shows a high potential for regulating low-temperature stress (Figure 5A). Transcriptome analysis confirmed the upregulation of *BrGeBP5* expression under low-temperature treatment, further validated using qRT-PCR. Promoter element analysis showed the presence of low-temperature corresponding elements, abscisic acid, and salicylic acid-responsive elements in *BrGeBP5*. Previous investigations have found salicylic acid to be effective in enhancing the growth of watermelon and citrus plants and cold tolerance in maize, potato, rice, and other plants under cold stress [21,22,23,24]. Additionally, low-temperature stress induces ABA production, which plays a vital role in enhancing plant stress resistance by promoting the activity of the antioxidant defense system and preventing oxidative stress [25]. Additionally, the prediction of target genes suggests that both Br-miR172d-5p and Br-miR5714 target *BrGeBP5*. Studies on the *A. thaliana miR172* have demonstrated its responsiveness to changes in environmental temperature, with low temperatures increasing miR172d expression, consequently reducing the plant’s low-temperature sensitivity [26].

*BrGeBP3* and *BrGeBP14* potentially respond to drought stress in *B. rapa*. Transcriptomic analysis showed upregulation of both genes under drought treatment. qRT-PCR results indicated peak expression at four and six hours post-stress, respectively, before declining. Promoter analysis revealed MeJA, gibberellin, abscisic acid, and auxin-responsive elements in both genes. Drought conditions prompt plants to reduce transpiration and avoid dehydration through certain mechanisms, like stomatal closure and canopy growth inhibition. Tomato studies have indicated that drought induction leads to GA inactivation, which causes early stomata closure during soil dehydration and inhibits leaf GA synthesis, thus, limiting the transpiration area [27]. Drought stress increases ABA content, leading to stomatal closure for water loss reduction, with ABA signaling being a core drought stress signaling pathway in plants [28]. Exogenous IAA improves drought tolerance in white clover [29]. *BrGeBP3* also contains a salicylic acid-responsive element. These findings provide important evidence for the role of GA, ABA, and IAA in drought stress adaptation. Protein interaction studies have shown that *At5g28040* (a homolog of *BrGeBP3* and *BrGeBP14*) interacts with DES1. H_2_S peroxidizes DES1 and promotes H_2_S production, which mediates peroxidative sulfation of OST1/SnRK2.6, which positively regulates ABA signaling and accelerates stomatal closure [30]. Additionally, drought-induced hormones (ABA, SA, MeJA, and ethylene) and ROS signaling vary among plant species, which promote H_2_S accumulation in guard cells, consequently initiating downstream signaling for stomatal closure and finally enhancing plant drought stress resistance [31].

## 4. Materials and Methods

### 4.1. Identification and Physicochemical Characterization of GeBP Family Genes

*A. thaliana* genome data were downloaded from TAIR and BRAD, and candidate *B. rapa GeBP* family members were searched by two-way BLAST in the *B. rapa* genome. This was followed by further identification of candidate genes using conserved domain analysis. The TBtools and Expasy (https://www.expasy.org/, accessed on 4 November 2022) software were used to analyze the molecular weight (MW), isoelectric point (pI), and other physicochemical properties of the *GeBP* family proteins in *B. rapa* [32,33].

### 4.2. Phylogenetic Relationships and Synteny Analysis

The gff3 file was downloaded from the *B. rapa* database, with the *GeBP* gene distribution on chromosomes being plotted using the TBtools software (v1.120). The phylogenetic trees of *GeBP* families in *B. rapa*, *O. sativa,* and *A. thaliana* were constructed by maximum likelihood estimation (MLE) using the MEGA-X software (v10.0.1) [34]. The phylogenetic tree was improved using iTOL (https://itol.embl.de/, accessed on 3 January 2023) to more clearly present the inter-species relationships [35].

### 4.3. Gene Structure and Analysis of Conserved Motif and Cis-Elements

The gene structure of *BrGeBP* was visualized using the TBtools Visual Gene Structure (basic) program. Its conserved motifs were analyzed using the MEME 5.4.1 online program (https://meme-suite.org/meme/tools/meme, accessed on 14 January 2023) [36]. The *B. rapa GeBP* was obtained through EnsemblPlants Family genes initiation codon 2000 bp upstream sequence, with the sequence analysis performed using the PlantCARE online website (http://bioinformatics.psb.ugent.be/webtools/plantcare/html/, accessed on 14 January 2023) [37,38].

### 4.4. Gene Expression Analysis

The transcriptome sequences of B. rapa of different tissues from NCBI GEO (https://www.ncbi.nlm.nih.gov/geo/, accessed on 11 November 2022) with the accession number GSE43245 were downloaded, and the data were normalized using the transcripts per million (TPM) method. The transcriptome sequences of *A. thaliana* under various abiotic stress treatments were downloaded, as shown in Appendix A. Gene expression profile heatmaps were prepared using TBtools (v1.120) [32].

### 4.5. Plant Material and Stress Treatments

*B. rapa* with stable self-incompatibility was utilized for the stress treatments. The plump seeds were seeded in MS modified medium (with vitamins, sucrose, and agar) and cultivated in a plant incubator. Seedlings with six leaves and similar growth status were selected for the stress treatments. The seedlings were placed in a hydroponic system with 150 mM NaCl to simulate salt stress and in 15% PEG6000 to simulate drought conditions. The plants were exposed to 4 °C for the cold stress treatment. We used unstressed *B. rapa* seedlings with the same growth period and under the same growth conditions as a control (CK). The duration of all stress treatments was 4, 6, and 12 h. Three biological replicates were run for each treatment group, and the samples were stored at −80 °C.

### 4.6. Total RNA Extraction and qRT-PCR

Total RNA was extracted using the SteadyPure Plant RNA Extraction Kit (Accurate Biotechnology, Hunan, China). These were then reverse-transcribed using the TransScript^®^ Uni All-in-One First-Strand cDNA Synthesis SuperMix for qPCR (TransGen, AU341-02, Beijing, China) for subsequent qRT-PCR analysis. The qRT-PCR reaction was performed on a qTOWER3 qPCR machine using the ChamQ SYBR qPCR Master Mix (Q711-03, Vazyme, Nanjing, China). The *BrActin2* was used as the internal reference gene, while relative expression level analysis of each gene was conducted using the 2^−ΔΔCT^ method. The gene-specific primer sequences are listed in Appendix A.

### 4.7. Statistical Analysis

The analysis of significant differences (a, b, c, d) was conducted using the single-factor ANOVA test on IBM SPSS Statistics 25 to compare the obtained means (with a = 0.05).

### 4.8. Prediction of Protein–Protein Interaction

For protein–protein interaction network analysis, predictions were made on the protein–protein interaction network prediction website (http://cn.string-db.org, accessed on 6 June 2023), where *B. rapa* was selected as the organism to obtain a protein–protein interaction map (minimum required interaction score = 0.400, using default settings for other parameters) [39].

### 4.9. Prediction of microRNAs Targeting BrGeBPs Genes

CDS sequences of *BrGeBP* were used to determine the interaction of genes with microRNAs using the psRNATarget database (http://plantgrn.noble.org/psRNATarget**,** accessed on 14 January 2023), and drawing was accomplished by using Excel.

## 5. Conclusions

In summary, we identified 20 *BrGeBPs* in the *B. rapa* genome. After the comprehensive analysis of sequence features, expression profiles, protein–protein interactions, prediction of microRNA targeted to *BrGeBPs,* and published data, we speculate that *BrGeBP5* has great potential in regulating low-temperature stress response, while *BrGeBP3* and *BrGeBP14* regulated the drought stress tolerance (Figure 8).

## Figures and Tables

**Figure 1 ijms-24-13597-f001:**
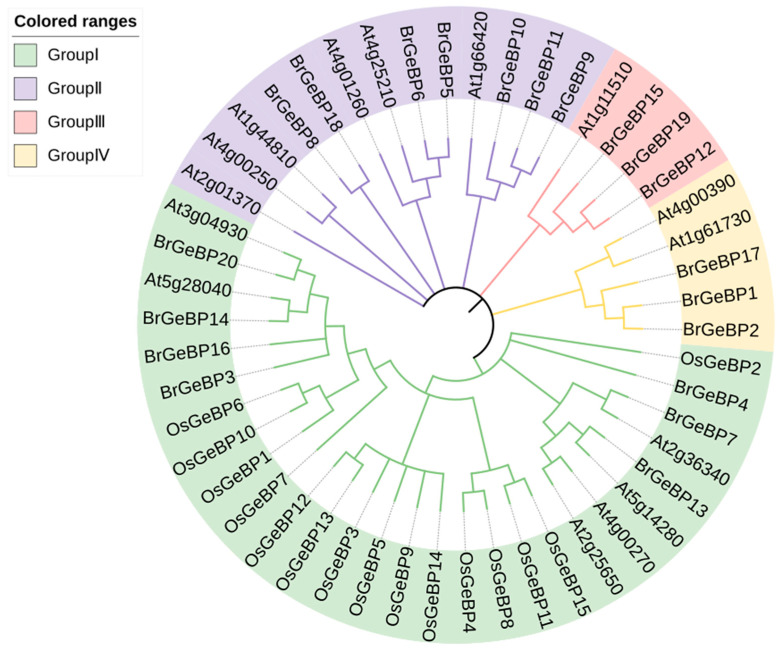
Evolutionary tree of *GeBP* family in *B. rapa, A. thaliana*, *and O. sativa*.

**Figure 2 ijms-24-13597-f002:**
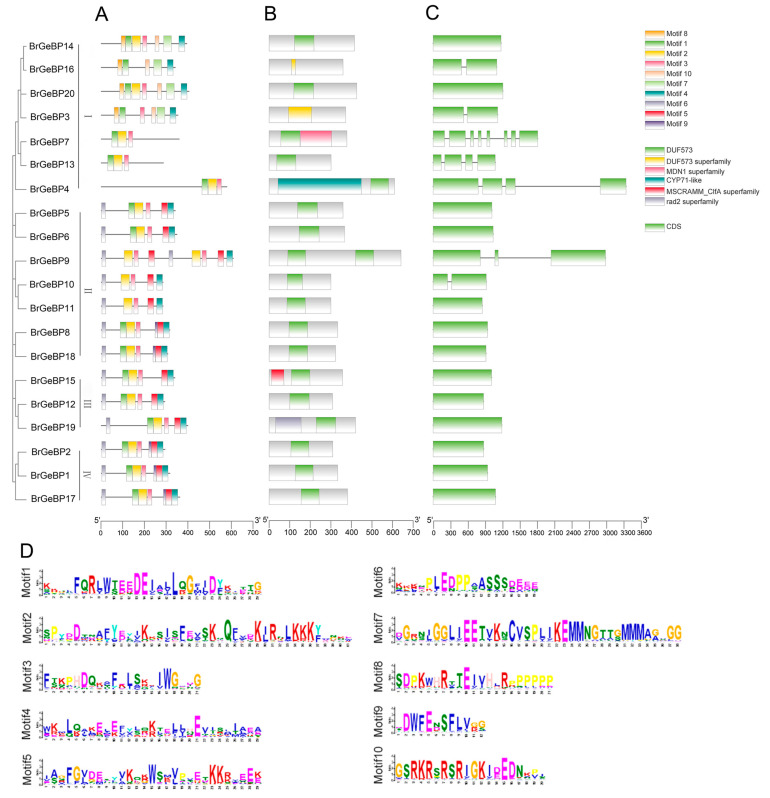
(**A**) Analysis of the conserved structural domains, (**B**) analysis of the functional structural domains, and (**C**) gene structure analysis of *BrGeBPs*. The exons and introns are represented by the green boxes and black lines, respectively. (**D**) Conserved sequence analysis of *BrGeBPs*.

**Figure 3 ijms-24-13597-f003:**
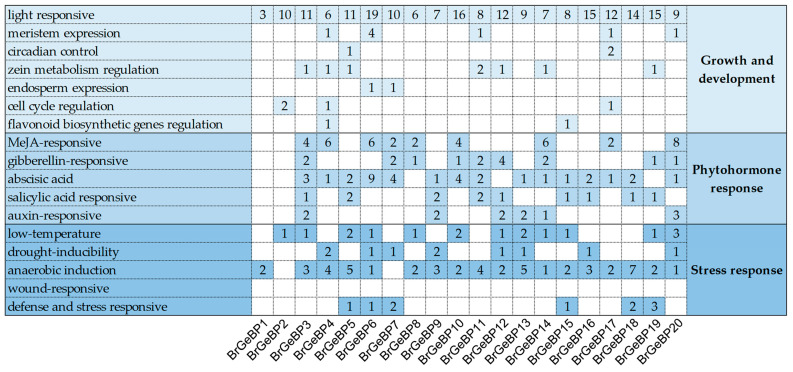
Analysis of *cis*-elements in promoter sequences of the *BrGeBPs*. The number of *cis*-elements in each gene is indicated by numbers.

**Figure 4 ijms-24-13597-f004:**
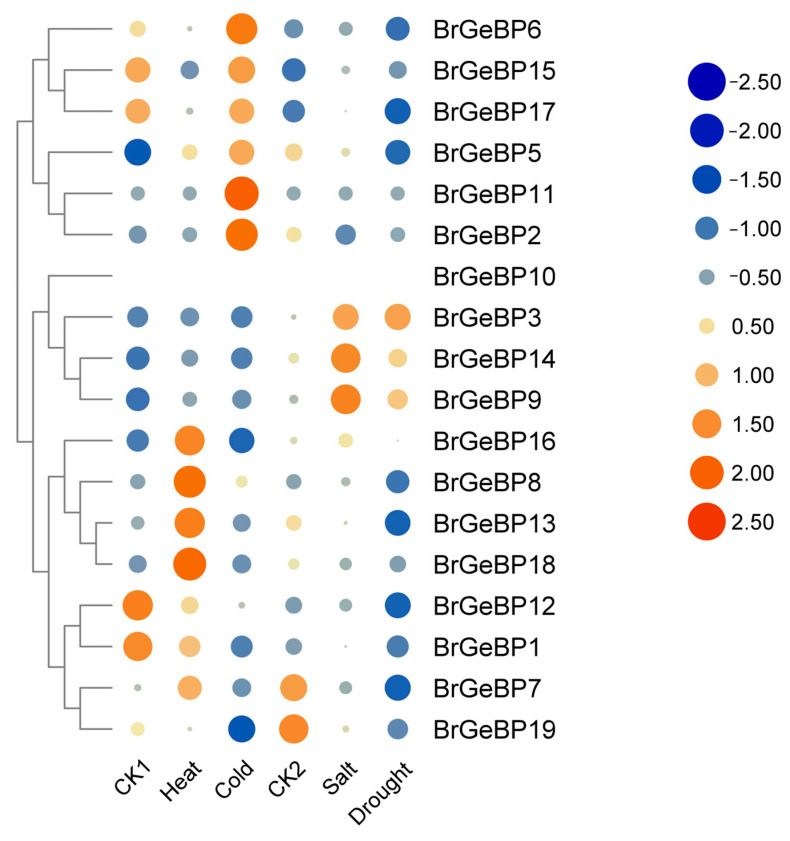
Expression pattern of *BrGeBP* genes in different organs and tissues of *B. rapa*. Darker red, larger dots indicate higher expression level; darker blue, larger dots indicate lower expression level.

**Figure 5 ijms-24-13597-f005:**
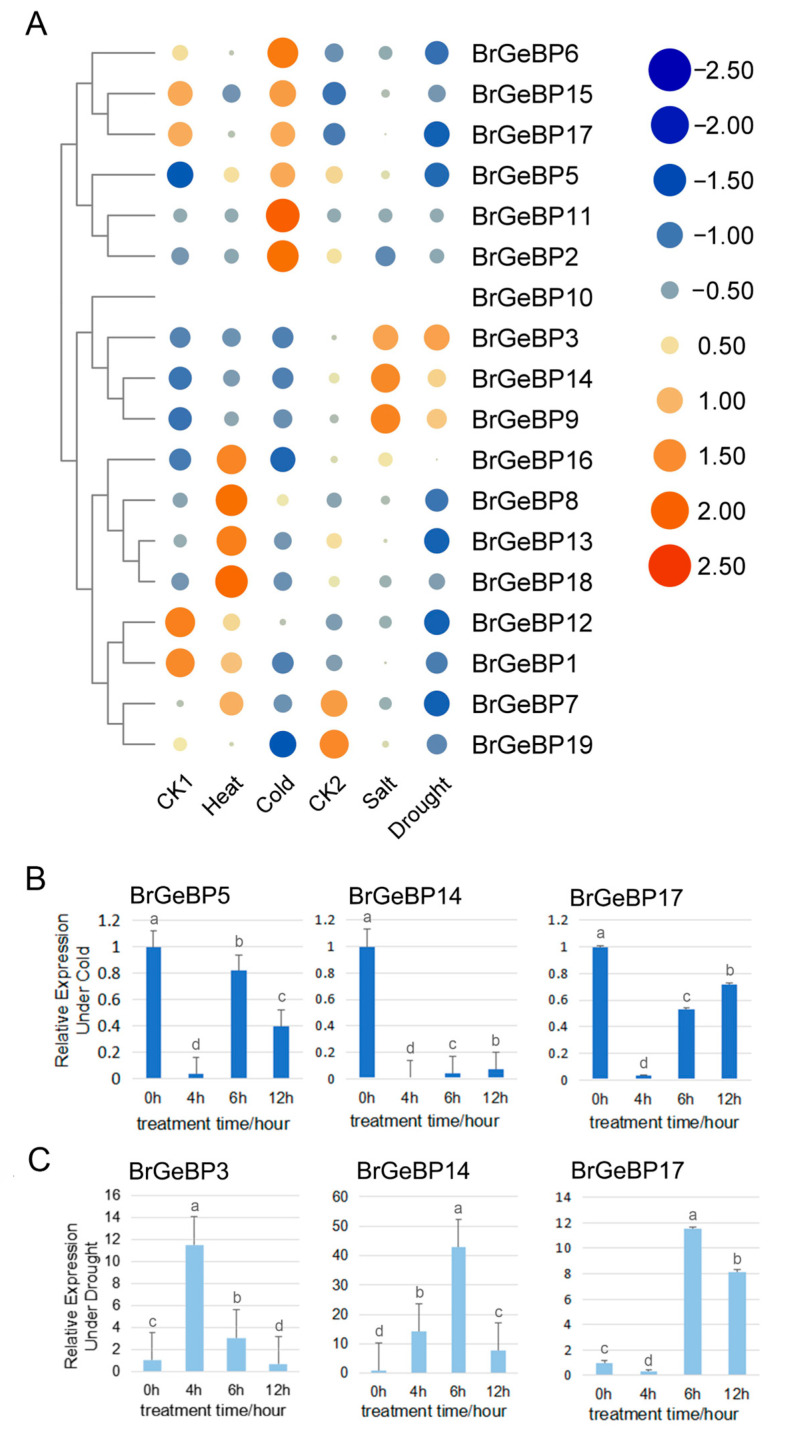
(**A**) Analysis of *BrGeBP* gene transcriptome data under abiotic stress. Darker red, larger dots indicate higher expression level; darker blue, larger dots indicate lower expression level; (**B**) cold treatment; (**C**) drought treatment. The above experiments were performed using 0 h as the control, with the treatment times set to 2, 4, 6, and 12 h. Each group had three biological replicates, with error bars indicating standard errors. Letters above the data bars indicate the statistical significance (the means are arranged in descending order, with the letter “a” after the highest mean, a = 0.05).

**Figure 6 ijms-24-13597-f006:**
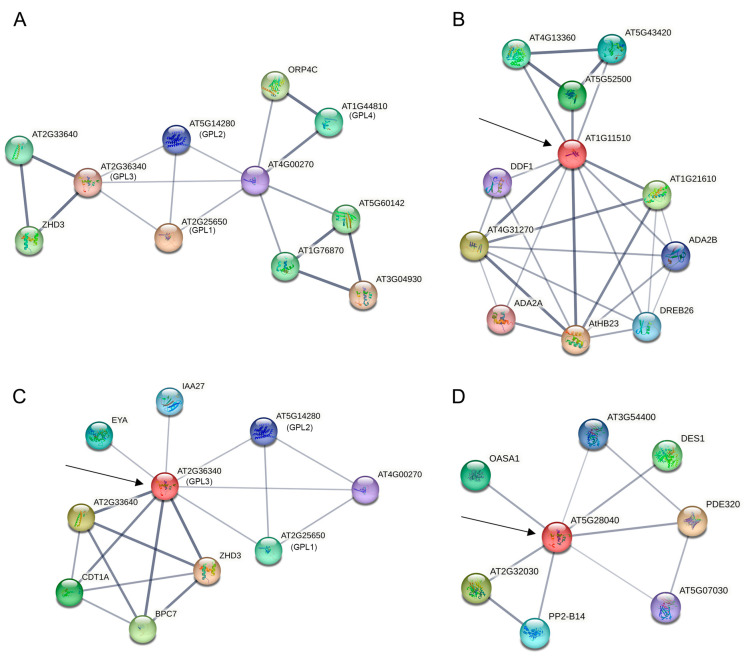
Protein–protein interaction networks of *GeBP* gene family in *A. thaliana*. (**A**) All *BrGeBP* PPIs; (**B**) *At1g11510* PPIs; (**C**) *At2g36340* PPIs; (**D**) *At5g28040* PPIs. Minimum required interaction score of 0.400; default settings were used for the other parameters. Network nodes represent proteins, edges represent protein–protein associations. Arrows indicate protein-interacting genes. The thinner the linkage and the darker the color, the stronger the correlation, and vice versa, the weaker the correlation.

**Figure 7 ijms-24-13597-f007:**
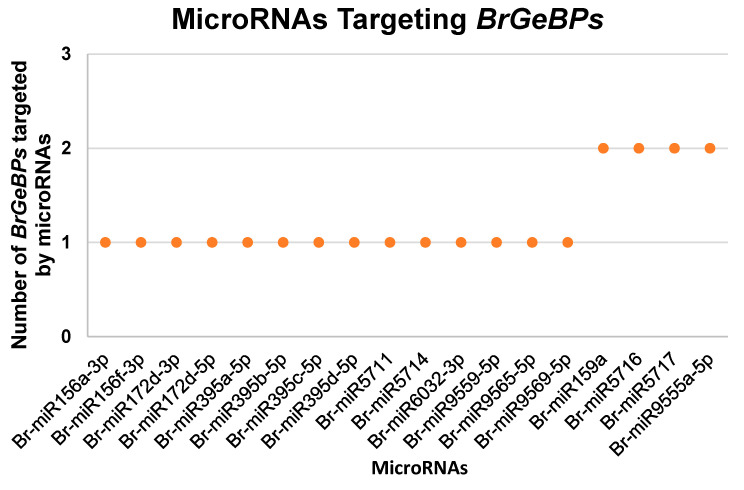
Smart graph illustration of microRNA targeting the *GeBP* genes in *B. rapa*.

**Figure 8 ijms-24-13597-f008:**
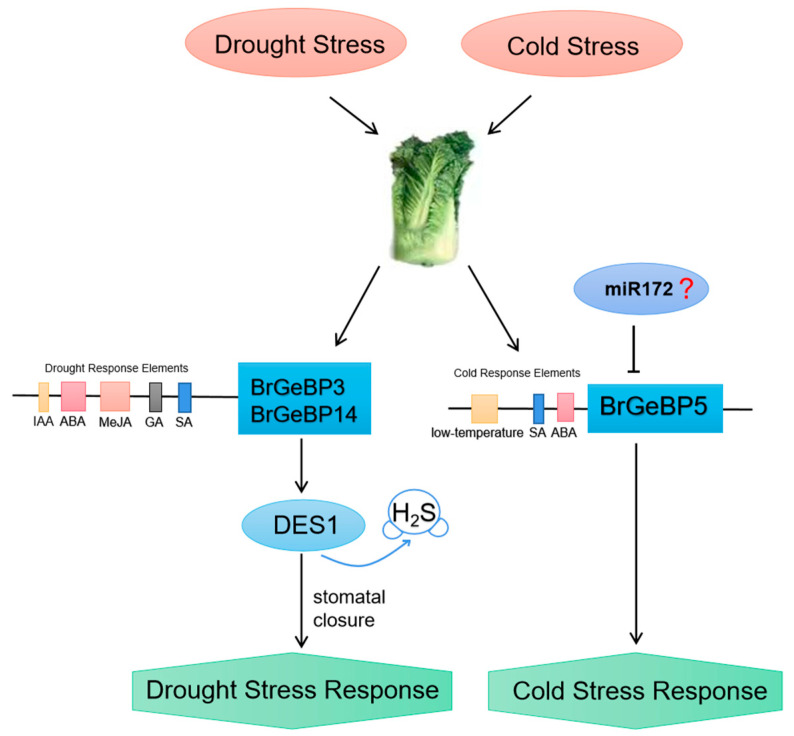
The putative molecular mechanism of *BrGeBPs*-regulated stress response.

**Table 1 ijms-24-13597-t001:** Information about *GeBP* gene family of *B. rapa*.

Gene ID	Gene Name	Chromosome (Chr)	Start	End	pI	Molecular Weight (Average)	Protein Length (aa)	Subcellular Location	*A. thaliana* Homologous GeBP Genes
Bra031402	*BrGeBP1*	A01	17755753	17756751	5.09	36,940.88	332	nucl: 13	*At4g00390*
Bra021317	*BrGeBP2*	A01	22344927	22345853	8.89	34,587.78	308	nucl: 14	*At4g00390*
Bra040116	*BrGeBP3*	A01	28356166	28357348	5.17	40,831.34	371	nucl: 8, cyto: 3, vacu: 2	*At5g28040*
Bra013842	*BrGeBP4*	A01	8032900	8036434	8.95	69,620.11	609	chlo: 9, nucl: 2, extr: 2	*At5g25140*
Bra013873	*BrGeBP5*	A01	8175027	8176103	7.53	39,311.39	358	nucl: 11, chlo: 1, cyto: 1	*At4g25210*
Bra019183	*BrGeBP6*	A03	25822483	25823583	5.72	40,527.59	366	nucl: 12, cyto: 1	*At4g25210*
Bra017244	*BrGeBP7*	A04	15774327	15776242	4.56	43,160.65	377	nucl: 10, cyto: 3	*At2g36340*
Bra032717	*BrGeBP8*	A04	5236288	5237286	5.31	36,619.28	332	nucl: 13	*At1g44810*
Bra028105	*BrGeBP9*	A04	5768566	5771723	8.98	72,816.9	640	nucl: 13	*At1g61730*
Bra028106	*BrGeBP10*	A04	5772763	5773737	6.17	33,634.76	299	nucl: 14	*At1g11510*
Bra028108	*BrGeBP11*	A04	5776855	5777754	5.91	33,917.13	299	nucl: 14	*At1g61730*
Bra025445	*BrGeBP12*	A04	8693888	8694811	5.61	34,648.02	307	nucl: 13	*At1g11510*
Bra039500	*BrGeBP13*	A05	9536224	9537362	5.71	34,161.25	300	nucl: 10, cyto: 2, plas: 2	*At5g14280*
Bra009986	*BrGeBP14*	A06	18469313	18470557	4.87	44,964.03	414	nucl: 12, cyto: 1	*At5g28040*
Bra019847	*BrGeBP15*	A06	4019905	4020975	7.66	39,168.47	356	nucl: 14	*At1g11510*
Bra036116	*BrGeBP16*	A09	2641324	2642491	4.69	38,400.59	358	nucl: 12, cysk: 1	*At5g28040*
Bra027078	*BrGeBP17*	A09	8253097	8254239	5.15	42,108.78	380	nucl: 13	*At1g61730*
Bra034488	*BrGeBP18*	Scaffold000096	356960	357928	5.35	35,634.26	322	nucl: 13	*At1g61730*
Bra038583	*BrGeBP19*	Scaffold000149	250721	251980	5.08	46,353.94	419	nucl: 10, cyto: 4	*At1g61730*
Bra040416	*BrGeBP20*	Scaffold000203	97200	98477	4.84	46,795.99	425	nucl: 13	*At3g04930*

aa, amino acids. Subcellular location: chlo (chloroplast), cyto (cytosol), nucl (nucleus), extr (extracellular), cysk (cytoskeleton), plas (plasma membrane), vacu (vacuolar membrane). The numerical values indicate the expression of each gene in different organelles.

**Table 2 ijms-24-13597-t002:** Details of *BrGeBPs* and targeted microRNAs.

MicroRNAs	MicroRNA Targeting *BrGeBPs*	
Br-miR156a-3p	*BrGeBP9*	
Br-miR156f-3p	*BrGeBP9*	
Br-miR172d-3p	*BrGeBP15*	
Br-miR172d-5p	*BrGeBP5*	
Br-miR395a-5p	*BrGeBP4*	
Br-miR395b-5p	*BrGeBP4*	
Br-miR395c-5p	*BrGeBP4*	
Br-miR395d-5p	*BrGeBP4*	
Br-miR5711	*BrGeBP19*	
Br-miR5714	*BrGeBP5*	
Br-miR6032-3p	*BrGeBP19*	
Br-miR9559-5p	*BrGeBP20*	
Br-miR9565-5p	*BrGeBP20*	
Br-miR9569-5p	*BrGeBP20*	
Br-miR159a	*BrGeBP8*	*BrGeBP18*
Br-miR5716	*BrGeBP9*	*BrGeBP10*
Br-miR5717	*BrGeBP4*	*BrGeBP13*
Br-miR9555a-5p	*BrGeBP13*	*BrGeBP7*

## Data Availability

All the data that support the findings of this study are available in the paper and its Appendix A published online.

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
