# Peer review of "Genome-Wide Identification and Expression Analysis of BrGeBP Genes Reveal Their Potential Roles in Cold and Drought Stress Tolerance in Brassica rapa"

_ijms, 2023, doi:10.3390/ijms241713597_

Round 1

Reviewer 1 Report

The manuscript entitled "Genome-Wide Identification and Expression Analysis of 2 BrGeBP Genes Reveal Their Potential Roles in Cold and 3 Drought Stress Tolerance in Brassica rapa" is well written. However, few major necessary questions needs to be addressed before publishing in reputed journal.

Authors screened the BrGeBP genes in several directions like development and stress. Several things making little confusion. So, authors are suggested to add one schematic diagram to explain the involvement of  BrGeBP genes in different pathways. Also for the stress, why authors have selected only cold and drought, explain? The expression of genes can be also investigated against high temperature, flood, metals etc.

In Figure 5.  No statistical significance given.

How, these genes are behaving with different growth conditions, difficult to understand without seedling growth pictures and parameters?

Reviewer 2 Report

Dear Authors 

I have read with interest the manuscript "Genome-Wide Identification and Expression Analysis of BrGeBP Genes Reveal Their Potential Roles in Cold and Drought Stress Tolerance in Brassica rapa". The article fits the journal's profile. The study is of a standard molecular biological nature. The authors identified the GeBP gene family in Brassica rapa, showed the chromosomal localization of the genes of this family. Analysis of the promoter regions of the genes allowed them to identify a number of cis-elements regulated by components of the hormonal signaling system and stress-dependent regulatory proteins. In contrast to other similar studies, the authors identified microRNAs that target individual members of the GeBP gene family. The novelty of this study relates to the first attempt to study the GeBP gene family in Brassica rapa.

            The weakest point of the manuscript is speculation about the possible functions of the genes studied, although some attempt was made to evaluate gene expression under conditions of osmotic stress, salinity, and cold. The authors' interpretation of the results on changes in gene expression raises questions. For example, it is stated that if certain analyzed genes change their expression levels in flowers, root system, and during growth and development, then "these findings indicate that BrGeBPs are crucial for organ development, fruit development, and sexual reproduction" (line 145-147) or "Conversely, under drought stress (Figure 6C), BrGeBP3 and BrGeBP14 showed higher expression levels than the control, peaking at 4 h, and 6 h. Furthermore, BrGeBP5 was significantly down-regulated after cold treatment, suggesting its involvement ment in cold stress, while BrGeBP3 and BrGeBP14 may play a key role in drought tolerance (lines 161- 165).

            There are some technical inaccuracies in the text. For example, for some reason the authors decided to call flowers, leaves, stem, etc. tissues ("The expression patterns of BrGeBPs in six tissues (callus, flower, leaf, root, silique, 133 and stem) were investigated based on public transcriptome data" (lines 133-e134).

            Polyethylene glycol causes osmotic stress and water deficit in plants, but not drought.

            I believe that the manuscript can be accepted for publication after a more rigorous presentation of the potential functions of the analyzed genes.

            Kind regards

Round 2

Reviewer 1 Report

The authors have incorporated all the suggestions. I recommend this manuscript to publish in the current version.